# Life-Cycle Assessment in Agri-Food Systems and the Wine Industry—A Circular Economy Perspective

**DOI:** 10.3390/foods14091553

**Published:** 2025-04-28

**Authors:** Catarina Marques, Sinem Güneş, Alice Vilela, Reinaldo Gomes

**Affiliations:** 1CITAB, Centre for the Research and Technology of Agro-Environmental and Biological Sciences and Inov4Agro, University of Trás-os-Montes and Alto Douro, 5000-801 Vila Real, Portugal; 2Graduate School of Natural and Applied Sciences, Environmental Engineering, Dokuz Eylül University, 35220 Izmir, Turkey; sinnemgunnes@gmail.com; 3CQ-VR, Chemistry Research Center, Department of Agronomy, School of Agrarian and Veterinary Sciences, University of Trás-os-Montes and Alto Douro, 5000-801 Vila Real, Portugal; avimoura@utad.pt; 4INESC TEC-Institute for Systems and Computer Engineering, Technology, and Science, Rua Dr. Roberto Frias, 4200-465 Porto, Portugal; reinaldo.s.gomes@inesctec.pt

**Keywords:** Life-Cycle Assessment (LCA), circular economy, sustainable food systems, environmental impact, resource optimization, waste valorization, renewable energy, agri-food sector

## Abstract

Rapid population growth, climate change, and resource depletion pose significant challenges to global food production, demanding sustainable solutions. A Life-Cycle Assessment (LCA) provides a structured framework for evaluating the environmental impact of food systems throughout their entire life cycle. This review examines how an LCA can be integrated with circular economy principles to address sustainability challenges, optimize resource use, and minimize waste in food and alcoholic beverage production. A systematic review of LCA applications in the agri-food sector was conducted, analyzing studies published across different regions. The selection criteria included relevance to circular economy strategies, waste valorization approaches, and assessing environmental impacts using LCA methodologies. The analysis explores explicitly the synergy between food and wine production within the broader agri-food system, considering shared sustainability challenges and opportunities for resource optimization. Key methodologies include cradle-to-grave assessments and the evaluation of waste-to-resource technologies. The findings demonstrate that LCA effectively identifies critical environmental hotspots, enabling the implementation of eco-design and resource recovery practices. Circular strategies, such as the use of renewable energy, precision agriculture, and nutrient recovery, significantly enhance sustainability. However, gaps remain in accounting for social and regional variability, as well as in integrating advanced technologies. When combined with circular economy principles, LCA provides actionable insights for policy development and sustainable practices. Advancing LCA methodologies and fostering multi-stakeholder collaboration are essential for creating resilient and environmentally responsible agri-food systems.

## 1. Introduction

Rapid population growth, climate change, resource depletion, and environmental degradation are increasing pressure on food production. These critical parameters have a direct impact on societies and their essential life-sustaining activities, including food production and the preservation of water availability [1]. The global food supply chain is one of the most intricate and interrelated systems on Earth. Also, it is one of the least sustainable supply chains [2]. Although, over millennia, ecosystems and human societies have adapted to relatively stable climate conditions, most agricultural practices are still affected by this pressure [1]. This results in numerous global food production challenges threatening the sustainability, security, and resilience of food systems worldwide.

Agri-food chains are intricate networks of activities and relationships connecting producers, processors, distributors, and end consumers. These are hierarchical organizations where producers are positioned at the bottom and final consumers at the top of the chain [3]. While agricultural producers confront increasing uncertainty due to anticipated environmental changes [4], food waste, biodiversity loss, and unequal resource distribution further complicate efforts to meet the growing demand for food. This complication depletes natural resources, pollutes the environment, and contributes to climate change. However, the challenges in food production alone are not sufficient to address the complexity of the global food supply chain. Issues of distribution, waste management, and access to resources are equally critical, requiring comprehensive solutions that span beyond the farm gate.

Food waste occurs at every stage of the food supply chain, from farming to the consumer level. It includes unharvested crops, trimmings and peels, by-products, overstocked or expired products, damaged goods, plate waste, unused ingredients, leftovers, and packaging waste. According to the European Environment Agency (EEA) [5], biowaste accounts for the most considerable portion (34%) of solid waste in the EU, with food waste comprising 60% of this category [2]. Food waste is a concern not only for the food itself, but also for the water, energy, and land resources involved in its production. It is estimated that approximately one-third of all food produced in the food chain is treated as waste, totaling over 1.3 billion tons annually [2,6]. This waste, compounded by distribution inefficiencies and resource loss during transportation, not only exacerbates environmental and economic burdens, but also underscores the urgency of addressing distributional imbalances. As a global issue, food waste has significant environmental and socioeconomic effects. The amount and type of food waste differs based on the country, economy, climate, culture, and location of food production and consumption [2]. Although supply-side strategies may increase per capita of food availability, they are approaching their limits. By 2050, the demand for agricultural products is expected to increase by 50% compared to current levels, while trends in crop yield growth are not promising for achieving the projected 60–70% increase needed to meet future food and fuel demands [1,6].

Agri-food chains are crucial to a region’s economy, as they generate added value and create jobs, serving as a key economic driver for rural and peripheral areas, which often underutilize their potential [3]. Also, biowaste is a significant European waste stream that can support the circular economy [2]. Although emerging technologies alone cannot resolve all challenges, innovative changes in food production, distribution, consumption, and disposal are crucial for achieving global sustainable food production [1]. Hence, moving towards integrated solutions that encompass the entire food system—production, distribution, and waste management—can provide a more holistic approach to sustainability. Thus, increasing the incorporation of sustainability considerations, focusing on promoting low-carbon food systems that integrate healthy and environmentally sustainable dietary practices, is needed [7]. This perspective prompts the food industry to acknowledge its responsibility in safeguarding the environment, leading many to seek enhanced sustainability practices [8] actively.

Developing sustainable and competitive agri-food chains helps reduce unemployment and stimulate local economic activities, providing small producers with access to new markets and enhancing their turnover and investment opportunities [3]. Achieving global food production is a complex challenge that requires more equitable food distribution systems, reduced food waste, strengthened regional foodsheds, and the correction of harmful market incentives, among other strategies [1]. Addressing distribution inefficiencies, such as uneven food access and supply chain waste, becomes equally crucial for ensuring a sustainable and equitable global food supply. While food serves as a carrier of both energy and water [7], addressing these challenges requires innovative solutions and a shift towards more sustainable practices, integrating circular economy principles and leveraging advancements in technology to ensure a stable and equitable food supply for future generations. The sustainability of the food supply chain entails implementing farming and management practices that safeguard the environment, assure workers’ health, and foster the economic development of local communities [3].

Additionally, intensive farming practices often lead to soil depletion, biodiversity loss, and pollution from the use of fertilizers and pesticides. Food production significantly contributes to environmental degradation through the release of greenhouse gases, deforestation, and excessive water consumption. Water is necessary for energy generation, and energy is required for the transportation and irrigation of agricultural fields [7]. As the interdependence between energy and water is essential for food cultivation and production, studies about evaluating the alterations in agricultural conditions resulting from climate change by examining shifts in climatic variables and their potential effects on crop yields are receiving attention [1,7].

Numerous methods and tools are currently available to evaluate the environmental impacts of agricultural production and food transformation. Life-cycle tools are highly effective in this context, enabling researchers to assess the ecological sustainability performance (Life-Cycle Assessment—LCA, Figure 1), economic sustainability (Life-Cycle Costing—LCC), and social sustainability performance (Social Life-Cycle Assessment—SLCA) [3]. These tools can be classified into qualitative methods, such as checklists and guidelines, and quantitative approaches, either monocriteria—like the Carbon Footprint—or multicriteria, such as a Life-Cycle Assessment (LCA) [8]. Among these tools, the LCA is a promising methodology for analyzing the environmental profile of various process schemes, and it is undoubtedly the one garnering the most attention from the international scientific community, as it is applicable across all sectors of the economy [6]. It is a frequently employed tool for analyzing environmental impacts and resource consumption throughout the entire life cycle of a product [9]. An LCA can help identify key actions for reducing a product’s environmental impact during the design of innovative products or the redesign of existing ones. This process, known as eco-design and eco-innovation, addresses various sustainability challenges [8].

The agri-food sector is one of the most actively engaged in LCA studies [3]. It is distinguished by unique characteristics that set it apart from other sectors, primarily because of the biological nature of its production processes [3]. The complete life cycle is examined from cradle to grave, encompassing the entire supply chain from raw materials to end-of-life waste treatment [11]. However, most food production LCA studies have defined system boundaries at the farm gate or production phase, often excluding the preparation, consumption, and waste management phases. This exclusion primarily results in greenhouse gas emissions (GHG emissions); therefore, addressing this data gap is essential for accurately assessing the environmental footprints of food products [12]. Additionally, to evaluate the cost-effectiveness of the proposed extraction methods, the operating costs of the entire production process need to be considered in food production [6].

At the farm level, an LCA systematically evaluates the inputs, such as fertilizers, water, and energy, as well as the outputs, including agricultural yields and waste products, associated with crop and livestock production, providing critical insights into the environmental impacts of various farming practices. Meanwhile, inputs and outputs of LCA can differ significantly based on the specific crop or livestock system [13].

By assessing the entire production chain—from resource extraction to waste management—an LCA plays a vital role in understanding how agricultural practices contribute to broader environmental challenges, such as climate change and resource depletion [13]. However, past LCA studies for agri-food products have significant diversity and are often challenging to compare. Thus, life-cycle thinking is crucial for facilitating the transition towards sustainability [8]. Considering this approach, identifying areas for improvement in sustainability and refining the formulation of policies and practices aims to mitigate the environmental footprint of agriculture [13].

This research highlights the critical role of the agri-food sector in environmental impact studies while addressing significant gaps in current LCA approaches. It contributes to the existing body of knowledge by extending system boundaries beyond the farm gate to include preparation, consumption, and waste management phases, providing a more comprehensive view of food production’s environmental footprint, particularly regarding greenhouse gas emissions, by its integration of an LCA with circular economy principles.

This innovative approach fills critical data gaps and offers actionable insights for improving sustainability practices in the agri-food sector, including the alcoholic beverages industry. It fosters an understanding of environmental impacts and informs future policy and practice.

## 2. Methodology

This review was conducted using a systematic approach to ensure the robustness and replicability of the findings. A comprehensive search was performed across multiple scientific databases, including Scopus, Web of Science, and Google Scholar. The search strategy included a combination of keywords related to an LCA, circular economy, sustainable food systems, resource optimization, and waste valorization. The keywords used were “Life Cycle Assessment in food production”, “circular economy in agriculture”, “sustainable food processing”, “valorization of agri-food waste”, and “resource optimization in agri-food chains”.

Inclusion and exclusion criteria were applied to refine the selection of relevant studies. Studies were included if they were peer-reviewed, published in English mainly within the last 5 years, and specifically addressed the integration of an LCA with circular economy principles in the agri-food sector. Exclusion criteria involved studies that lacked quantitative data, review papers without methodological rigor, and research focused solely on industrial or non-food-related applications of LCAs.

A total of 174 studies were initially identified, of which 102 studies met all inclusion criteria and were selected for detailed analysis. This process ensured that the review captured a diverse and representative sample of the existing literature while maintaining methodological rigor.

## 3. LCA and Circular Economy in Food Production

The global food industry has become an increasingly focal point for discussions on environmental sustainability [14]. LCA applications in food production have become an international priority to meet United Nations-announced sustainable development goals (SDGs) [8]. The LCA framework outlined by the Society of Environmental Toxicology and Chemistry (SETAC) [15] offers a structured approach to evaluating environmental impacts. As shown in Table 1, the process begins with goal definition and scoping, where the assessment’s objectives, system boundaries, and functional units are established. This is the first phase, followed by inventory analysis, which involves gathering and quantifying data on resource use, emissions, and waste generated throughout the product’s or process’ life cycle. The next phase, the impact assessment, is divided into three key steps: classification, where the inventory data are assigned to relevant environmental impact categories; characterization, which quantifies the extent of these impacts; and valuation, where the significance of each effect is evaluated. The final phase, improvement analysis, focuses on identifying strategies to reduce the most significant environmental impacts, enabling the implementation of more sustainable practices. This comprehensive structure ensures a detailed examination of a product’s environmental footprint and guides the development of environmentally conscious improvements. In Europe, food production is responsible for 20–30% of all anthropogenic environmental impacts [8]. However, resource-intensive processes, ranging from agricultural practices to distribution networks, significantly contribute to environmental challenges.

To address these challenges, LCAs and circular economy concepts have emerged as vital tools for quantifying environmental impacts and proposing sustainable solutions [16]. Research on forecasting LCA impacts has expanded with the integration of models such as system dynamics modeling and agent-based modeling. System dynamics enable the simulation of complex interactions among agricultural production, resource consumption, and waste management, allowing for the prediction of long-term environmental impacts across the entire food production system. Agent-based modeling, on the other hand, helps assess how individual decision makers, such as farmers, distributors, and consumers, affect the flow of materials and resources, offering insight into how different stakeholders can contribute to reducing the environmental footprint of the food supply chain. By incorporating these advanced forecasting models, we can gain a deeper understanding of the cascading effects of food production and waste management, enabling more accurate projections of the impacts of various circular economy strategies. To achieve the aims of the global food system, studies must account for the known trade-offs between agricultural production and ecological goals, such as water and energy conservation, biodiversity and habitat protection, and reductions in greenhouse gas (GHG) emissions [1]. The most widely adopted approach involves assessing environmental impacts relative to the quantity of product output [14]. The proper management of both upstream farming practices and downstream product production is crucial for ensuring the development of high-quality products that are competitive on an international scale [17]. While LCA has proven instrumental in evaluating the environmental impacts of food production, gaps remain in fully understanding the long-term effects of sustainable practices, mainly when applied across different geographical regions and production scales [16].

Recent reviews in the agri-food sector increasingly emphasize the value of integrating multiple life-cycle approaches. Table 2 summarizes a comparative overview of key studies applying an LCA, SLCA, and LCC in agri-food systems. For instance, SLCA studies evaluating coffee and cocoa production highlight significant differences in social outcomes based on the farm size, wage structures, and regional labor conditions [18]. Economically, LCC approaches in olive oil production demonstrate that valorizing by-products (e.g., for energy or feed) can reduce total system costs by up to 25% compared to linear waste disposal models [19]. This quantitative synthesis shows that incorporating social and economic dimensions enhances the capacity of LCA frameworks to capture the multifaceted impacts of food production.

In contrast, a circular economy seeks to establish a closed-loop system that minimizes waste and optimizes the resource efficiency by recycling, reusing, and repurposing materials throughout the entire supply chain.

The circular economy is guided by three core principles [16,20] that encourage maximizing the value of resources, preserving natural capital, and enhancing system effectiveness. As detailed in Table 3, applying these principles in food production places a strong emphasis on efficient resource use, waste reduction, and environmental preservation.

While these principles provide a clear roadmap for sustainability in food production, several barriers exist to implementing circular economy models in the agricultural and food sectors. One significant barrier is the financial cost associated with transitioning from linear to circular models, particularly for small-scale producers who may lack the resources to invest in new technologies or processes. Additionally, there are infrastructural challenges, such as the lack of facilities for recycling or reusing materials in certain regions, which can hinder the adoption of circular economy practices. Large producers, on the other hand, may face resistance to change due to established, resource-intensive practices and concerns about the economic viability of circular strategies in the short term.

The lack of knowledge and awareness about circular economy principles also poses a barrier. Many stakeholders in the agri-food sector, including farmers, processors, and consumers, may not fully understand the long-term benefits of implementing circular economy practices, such as improved resource efficiency and reduced environmental impacts. This knowledge gap can lead to reluctance to adopt new practices or technologies.

Additionally, policy and regulatory challenges play a role in slowing the transition. There is a need for clear, consistent policies that support circular economy initiatives and incentivize stakeholders to adopt more sustainable practices. Without the proper regulatory framework, businesses may have little incentive to shift from traditional linear models to circular approaches, particularly if the upfront costs are perceived as too high.

The first principle focuses on maximizing the utility of products, co-products, and by-products at every stage within and across supply chains, ensuring that materials retain their highest value [21]. This involves nutrient recovery and efficient resource management throughout the production stages.

The second principle supports preserving and enhancing natural capital by substituting finite resources with renewable ones [24]. This shift can be achieved in food systems through regenerative farming methods, which help sustain natural ecosystems and reduce dependency on limited resources.

The third principle promotes system effectiveness by targeting and reducing negative externalities, such as pollution and waste [25]. By fostering circular innovation, food production can become more sustainable and resilient, leading to a balanced and environmentally conscious system.

An LCA supports environmental policy development by providing objective, reliable, and comparable data [6]. Integrating an LCA into a circular economy framework presents a promising approach to enhancing sustainability in food production. By integrating an LCA into a circular economy model, food producers can identify key areas of inefficiency and environmental impact, enabling them to adopt more sustainable farming practices, enhance energy management, and effectively recycle waste [16,31]. A circular economy in food systems encourages producers to rethink conventional linear production models that end in waste generation.

Prioritizing low-carbon food patterns has become increasingly important, so the theoretical framework of the energy–water–food security (EWFs) nexus is employed to analyze the inter-relationship between energy and water use in food production [32]. The interdependence among these dimensions is evident: energy and water are essential for growing and producing food, while food, in turn, transports energy and water. Simultaneously, water is necessary for energy production, and energy is required for the transportation and irrigation of crops [7].

Numerous studies have examined the environmental impacts of coffee production and explored potential improvements that can be made while maintaining high-quality output. Most of this research has focused on the farming component of the production system, often exploring alternative inputs (e.g., organic versus inorganic fertilizers), the utilization of waste (e.g., for bioenergy), and the effects of co-producing other agricultural products [16]. An LCA’s approach aligns well with the food production industry, including wine, a significant economic stream in most Mediterranean countries. In wine production, substantial amounts of water, energy, and materials are consumed, and large quantities of waste are generated at various stages. For example, in wine production, grape pomace—a by-product of winemaking—can be repurposed into bioenergy or animal feed, reducing waste while adding economic value. Energy management strategies, such as adopting renewable energy sources in production facilities and utilizing sustainable packaging materials like lightweight glass or PET bottles, can further reduce the industry’s environmental footprint.

As another internationally traded product, coffee has significant socio-economic and environmental impacts [17]. The most important environmental impact of coffee production is its effect on land use, which covers approximately 1.2 million hectares [33]. Additionally, the use of chemical and organic fertilizers affects carbon emissions in coffee production [17]. On the other hand, cacao is a crucial export for many tropical countries, with approximately 5 million tons produced globally in 2022/2023 [34]. While cacao is traditionally grown under forest canopies or agroforestry systems, full-sun monocultures are increasingly promoted, relying on synthetic inputs and machinery, heightening the environmental impact [35].

A comparative analysis of circular economy implementation across different agri-food production methods reveals notable variations in both environmental and socio-economic outcomes. For example, integrated farming systems typically demonstrate higher resource efficiency and circularity scores compared to conventional monocultures, with a 20–30% increase in nutrient recycling rates [36]. Organic agriculture, while often associated with reduced yields, shows an enhanced performance in terms of social consequences, including local employment generation and improved community resilience. According to recent Eurostat data [37], regions with higher adoption rates of agri-food circular practices have also reported a 15% increase in rural employment. Such comparisons underline the importance of aligning circular strategies not only with environmental objectives, but also with social equity and economic development goals.

Beyond environmental sustainability, it is crucial to integrate social and economic dimensions into the evaluation of agricultural production systems. A Social Life-Cycle Assessment (SLCA) complements an environmental LCA by assessing the impacts of food production on the workers, local communities, and consumers. A recent systematic review by Bhatnagar et al. [18] highlights the role of an SLCA in facilitating the transition to a circular economy, emphasizing how social impact assessments contribute to more sustainable production systems. This study highlights the importance of integrating social considerations into life-cycle methodologies to ensure that sustainability efforts address ethical concerns, promote fair labor practices, and support community well-being.

Economic Life-Cycle Costing (LCC) further enhances this assessment by analyzing the financial feasibility of sustainable practices. Economic viability is influenced by market incentives, policy support, and consumer behavior, all of which affect the adoption of circular economy principles. In this context, Romero-Perdomo and González-Curbelo [19] emphasize the importance of integrating multi-criteria decision-making techniques into life-cycle assessment tools to support the transition to a circular bioeconomy. Their review demonstrates how combining environmental, social, and economic assessments can improve decision making in the valorization of agri-food waste biomass, contributing to more sustainable agricultural production systems. Collectively, these methodologies offer a comprehensive understanding of sustainability in the agri-food sector.

## 4. Farm-Level LCA: Inputs, Outputs, and Environmental Impact

At the farm level, LCA examines the inputs and outputs related to crop and livestock production, providing insights into the environmental impacts of agricultural practices [16]. The assessment of inputs in farming typically includes fertilizers, water, and energy, which are crucial for achieving optimal yields [7].

Fertilizers, though essential for enhancing soil health and productivity, can lead to nutrient runoffs, contributing to water quality degradation. In recent years, water use, particularly in regions prone to scarcity, has become increasingly critical, and managing this resource effectively helps minimize environmental stress [17]. Energy inputs, primarily fossil fuels, are necessary for operations like tillage, irrigation, and product transportation, which can significantly impact the farm’s ecological footprint. The efficient management of these resources, including the adoption of renewable energy, is a crucial aspect of sustainable farming systems [7,17].

Outputs, on the other hand, include primary yields and several forms of waste products, such as emissions, nutrient runoff, or excess water usage [9]. Technically, high yields often reflect productivity, yet they must be balanced against waste generation, such as unused crop residues and by-products. These wastes pose management challenges but also offer opportunities for resource recovery, such as transforming organic waste into biogas or utilizing crop residues as animal feed [11]. Such circular approaches enhance farm-level sustainability by reducing waste and creating additional value from byproducts, demonstrating the importance of integrating waste management strategies into farming systems [13,17].

The link between energy and water use in food production is vital. Livestock farming, for instance, contributes significantly to methane emissions through the process of enteric fermentation. At the same time, crop production can result in substantial CO₂ and nitrous oxide emissions due to the use of fertilizers and poor soil management practices [38]. Additionally, unsustainable water use in irrigation leads to resource depletion and harms ecosystems, particularly in areas prone to drought. Land degradation, resulting from over-cultivation and deforestation, compromises soil health and agricultural productivity, stressing the need for conservation-focused farm practices [38,39]. Agriculture is a significant contributor to greenhouse gas (GHG) emissions; GHGs trap heat in the Earth’s atmosphere, contributing to the greenhouse effect and driving global warming and climate change. Key greenhouse gases (GHGs), such as dissolved organic carbon (DOC), methane (CH_4_), and nitrous oxide (N_2_O), are released through activities including fossil fuel combustion, agricultural practices, industrial processes, and waste management. The farm size affects agrarian production inputs, which in turn impact GHG emissions; however, the specific effects and mechanisms involved remain unclear [13].

When agricultural production is scaled up, the environmental and economic outcomes are influenced not only by the increase in input demands, but also by regional characteristics such as the climate, soil type, and resource availability, as well as the degree of technological advancement. Larger farms in developed regions often benefit from precision agriculture and advanced waste management systems, enabling greater efficiency and the integration of circular economy technologies [13,17]. For instance, anaerobic digestion systems and composting facilities are more feasible at scale and can transform organic waste into energy or fertilizer, closing nutrient loops [7,31]. However, in less industrialized contexts, scaling may lead to higher emissions and resource inefficiencies due to limited access to such technologies [22]. Therefore, the effectiveness of circular economy strategies in scaled systems is highly dependent on regional infrastructure and policy support. This reinforces the importance of region-specific LCA studies that account for variability in production systems and technological capabilities [14].

Minimizing resource use and waste generation in agriculture requires implementing various strategies. Techniques like precision agriculture optimize the use of inputs such as water, fertilizers, and pesticides, reducing resource waste [11]. With practices such as reduced tillage and integrated pest management, conservation agriculture helps maintain soil health while decreasing chemical use. Livestock systems benefit from efficient waste management practices, such as improved manure storage and slurry acidification, which reduce emissions of harmful gases. By integrating livestock with crop production, mixed farming systems can improve nutrient recycling and the overall resource efficiency [17,40].

Circular economy approaches in food processing are particularly effective for waste valorization, transforming food waste into valuable products like bioenergy or compost. For example, the anaerobic digestion of food waste can produce biogas, which powers food processing plants, reducing reliance on external energy supplies. The digestate from this process can also be used as a biofertilizer, closing the nutrient loop and contributing to sustainable agriculture. An LCA plays a critical role in evaluating these valorization processes by quantifying the reductions in emissions, resource consumption, and waste disposal costs, supporting sustainable agricultural and processing practices [41].

Scaling up agricultural production while maintaining circularity introduces challenges that vary across regions due to differences in technology access, infrastructure, and socio-economic conditions. In developed regions such as the EU, circular technologies like anaerobic digestion, precision farming, and closed-loop irrigation systems are increasingly integrated into large-scale operations [42]. In contrast, smallholder systems in developing countries often rely on low-tech solutions such as composting and intercropping to close resource loops. These regional differences necessitate tailored circular economy approaches, supported by localized innovation and policy frameworks. Furthermore, when upscaled, environmentally friendly technologies may require complementary infrastructure—such as decentralized energy systems or logistics for secondary resource collection—to retain their circular benefits at a macro scale [43].

Evaluating inputs and outputs enables the identification of key areas where resource efficiency can be improved, waste generation can be reduced, and environmental impacts can be minimized. As the global food demand continues to rise amid growing environmental challenges, using an LCA to guide sustainable agricultural practices will be instrumental in shaping policy and informing decisions that promote a balance between productivity and environmental conservation [40,44].

## 5. LCA in Food Processing: Efficiency and Sustainability

Food processing is a resource-intensive stage in the food supply chain, characterized by substantial energy consumption, significant water usage, and substantial waste generation. At the same time, the rapid development of environmental issues has underscored the need to integrate sustainable food-processing practices [11]. An LCA enables the assessment of these processes, helping to identify inefficiencies and opportunities for improvements in sustainability.

The ISO 22000 [45] standard establishes a comprehensive framework for advancing and assessing food safety management systems. This standard sets parameters for efficient communication, quality assessment systems, prerequisite programs, and Hazard Analysis and Critical Control Points (HACCP) principles. Furthermore, implementing ISO 22000 fosters a culture of continuous improvement, enabling organizations to adapt to emerging challenges in food safety and align with global best practices.

When performing LCAs with extended time horizons and varying temporal scopes, it becomes pertinent to account for potential changes in characterization factors. Although this topic is frequently addressed in future-oriented (prospective) LCAs, only a few academic studies have thoroughly examined this issue [11]. By focusing on the environmental impacts of food processing, an LCA assesses everything from energy usage during food transformation to water consumption in cleaning and production lines. This comprehensive evaluation offers insights into the environmental impact of food processing and provides strategies for enhancing the resource efficiency and reducing waste [31].

An LCA plays a pivotal role in assessing the environmental performance of food processing systems by evaluating key metrics, including energy use, water consumption, and waste generation. In these systems, the energy footprint is significant due to the extensive use of electricity and thermal energy in processing, packaging, and distribution. For example, increasing the farm size reduces agricultural GHG emissions, as larger farms have higher fertilizer use efficiency [13]. On the other hand, water usage is another critical factor, as it is consumed in food production, cleaning, sanitization, and cooling processes. Waste generated, including solid waste, wastewater, and emissions, contributes to environmental degradation. An LCA provides a thorough analysis of these inputs and outputs, helping to identify the stages that have the most significant impact and formulate strategies to reduce energy consumption, conserve water, and manage waste efficiently [41].

Efficient measures, such as adopting renewable energy sources and optimizing production processes, are crucial for reducing the environmental footprint of food processing. In developing countries, particularly those with a predominantly smallholder agricultural sector, larger farms have the potential to reduce greenhouse gas (GHG) emissions, provided that the chemical use efficiency is improved [13]. In addition, growing environmental awareness has driven the pursuit of new extraction technologies that offer enhanced environmental profiles and greater efficiency, often referred to as “green emerging technologies” [6]. A circular economy approach in food processing involves recovering and valorizing waste streams, such as converting food by-products into energy, feed, or other valuable products. This aligns with zero-waste strategies, contributing to environmental and economic sustainability. By focusing on the recovery of resources, food processing plants can transform waste into valuable inputs, thus minimizing the environmental impact while supporting sustainable development [31,41].

Integrating renewable energy sources and optimizing processes within the sustainable food industry is crucial for reducing dependence on fossil fuels, minimizing environmental impacts, and enhancing operational efficiencies. Process optimization involves enhancing energy efficiency and minimizing waste through technologies such as heat recovery systems, energy-efficient equipment, and automation. Renewable energy sources, such as solar, wind, and bioenergy, can also be integrated into food processing operations, decreasing greenhouse gas emissions and reducing energy costs. LCA studies often highlight the benefits of renewable energy in reducing the carbon footprint of food production, making it an essential tool for sustainable development in this sector [29].

Recent studies have broadened the understanding of circular economy implementation by integrating social and behavioral dimensions alongside policy and organizational mechanisms. For instance, the research of Ruiz-Real and coworkers [46] on the social economy’s role in circular transition emphasizes the importance of participatory governance. Additionally, Yadav et al. [47] illustrate how circular strategies contribute to achieving the Sustainable Development Goals through regional cooperation. Other studies [48,49] approached consumer behavior, psychological enablers, and organizational circularity indicators. These perspectives complement technical and environmental analyses by offering a holistic view of the circular transition. Notably, European policymaking in this domain has increasingly focused on integrating such multi-dimensional frameworks to guide sustainability transformations [50].

Ultimately, LCA applications within circular economy frameworks emphasize the importance of resource efficiency in food processing. By converting waste into energy and other valuable products, the food processing industry can significantly reduce its environmental impact while enhancing the economic viability. For instance, water recovered from wastewater treatment processes can reduce water consumption in food plants. These resource recovery efforts conserve natural resources and contribute to the long-term sustainability of food production systems. An LCA offers the tools to assess these trade-offs, aiding decision making by prioritizing environmental sustainability and resource conservation [31,41].

## 6. LCA in Wine and Fermented Beverages Production

An LCA serves as a vital framework for examining the environmental impacts associated with the production of fermented beverages. This comprehensive method evaluates the entire life cycle of a product, which includes stages such as raw material extraction, production, distribution, and disposal. By doing so, an LCA provides valuable insights into sustainability and helps identify opportunities for improvement. In recent years, LCAs have been increasingly utilized to evaluate the environmental performance, sustainability, and overall efficiency of the wine industry. Notably, research indicates that production strategies have a more significant influence on the carbon footprint of wine production than the type of wine or grape variety used [51]. The main ecological hotspots identified are the viticulture phase due to the use of fuel, fertilizers, pesticides, the fermentation process, and the production of primary packaging, particularly glass bottles [44,52,53]. Electricity consumption, sugar usage, and liquid CO_2_ production are also significant contributors to environmental impacts, particularly in categories related to toxicity [54]. However, there is variability in system boundaries and a lack of site-specific data, which affects the consistency of results across studies [53].

The fermentation phase is a significant contributor to the environmental impact of producing fermented beverages. For instance, in the production of lactic acid bacteria concentrates, fermentation has emerged as a key phase, yielding significantly higher yields that result in reduced environmental impacts [52]. Similarly, in cellulase production, the fermentation phase was crucial, with solid-state fermentation (SSF) showing lower environmental impacts compared to submerged fermentation (SmF) [55].

An LCA enables producers to implement strategies that reduce environmental impacts by pinpointing critical stages and processes. Examples include using lighter glass bottles, composting biomass, and adopting organic grape production, which has been shown to improve the environmental performance [56]. Regarding the reduction of greenhouse gas emissions, the LCA emphasizes the importance of production strategies over the wine typology or grape variety. This includes optimizing production processes and adopting sustainable practices [51]. Quantifying material and energy consumption enables producers to maximize resource utilization and minimize waste. This is crucial for small winegrowers, who often have higher emissions than larger industrial producers [57].

LCA studies highlight energy use and packaging as significant contributors to the environmental impact of beer. The studies suggest that adopting efficient technologies, using recycled materials for packaging, and implementing sustainable agricultural practices can reduce the ecological footprint. However, the diversity in methodologies and impact categories used in studies complicates the synthesis of results [58,59].

Packaging and producing whisky, grain alcohol, and cereal are significant environmental burdens in the spirits sector. Strategies such as reusing or recycling packaging materials can help mitigate these impacts. The environmental effects of the manufacturing process are relatively limited but contribute significantly to the product’s life-cycle cost [60].

A significant challenge in applying an LCA to fermented beverages is the variability in system boundaries and methodologies, which makes it difficult to compare results across different studies. To improve the reliability of LCA results, more standardized approaches and comprehensive, site-specific data are needed [53,58]. To enhance sustainability, the industry can focus on improving the energy efficiency, adopting renewable energy sources, and optimizing packaging materials. For instance, utilizing recycled materials and implementing improved waste management practices can significantly reduce environmental impacts [58,60].

The development of standardized LCA procedures is crucial for enhancing the comparability and reliability of studies in the wine sector. Establishing international guidelines would help LCA practitioners and stakeholders implement more consistent and effective environmental assessments [61].

While LCAs have been extensively applied to wine and beer, there is limited research on other fermented beverages, such as spirits and fruit-based drinks. Expanding LCA studies to these areas can provide a more comprehensive understanding of the environmental impacts across the entire fermented beverages industry [62,63].

In conclusion, an LCA is a valuable tool for identifying environmental hotspots and guiding sustainability improvements in the fermented beverages industry. However, addressing data variability and methodological differences is crucial for enhancing the accuracy and comparability of LCA studies.

## 7. LCA of Packaging and Distribution: Closing the Loop

Urbanization has significantly increased waste generation within modern communities, driven by industrial production, the growth of the service sector, construction activities, and changes in human lifestyles [64]. Agri-food chains represent intricate networks of activities and interactions that connect producers, processors, distributors, and end consumers. Failing to address this issue will result in irreversible damage to both natural and human-made ecosystems [3,65]. When considering the demand volume, flow, and the impacts of production and distribution processes, the food industry emerges as a major contributor to climate change. This excessive waste presents significant environmental challenges, as a large portion of the nearly 2 billion tons of solid waste produced annually is poorly managed [66]. Furthermore, improper waste management can lead to long-lasting and irreversible environmental issues, while efforts to mitigate its environmental impacts may compromise the core principles of sustainable development. An LCA provides insight into the interconnections and feedback loops between the nexus and other environmental impact categories, considering the entire process from ‘cradle to grave’ [35]. Plastic packaging is the primary application by segment, followed by the building and construction sector in the EU market. However, OECD (2022) results show that only 9% of global plastic waste is recycled, while 22% is mismanaged [67].

However, pathways to mitigate these impacts exist, with reusable packaging networks offering timely solutions. Recycling methods can be classified into primary (to polymers), secondary (to polymers or monomers), tertiary (to naphtha or feedstock), and quaternary (to energy). While primary recycling is typically the most environmentally preferable, it is less tolerant of contaminants [68]. Despite the advantage of reducing virgin plastic polymers in food packaging, adopting reusable systems remains constrained by organizational and economic challenges. Packaging and containers, though unavoidable, are significant environmental stressors and primary sources of waste [69]. At the same time, reasonable concerns regarding their actual environmental benefits persist [69,70].

Moreover, recycling could be divided into two categories: closed-loop and open-loop. One approach involves recycling materials into the same products, while another focuses on creating different materials or products, often involving various stakeholders. Compared to technological challenges, the strategic or tactical issues in plastic waste management are more prominent and play a critical role in environmental mitigation [71]. The priority of the LCA criteria, such as accuracy, ease of communication, reproducibility, and explicitness, can vary depending on the context [72], influencing the choice of LCA methodology and often leading to trade-offs between these criteria. Life-cycle thinking and the LCA are essential tools for sustainable transitions, playing a significant role in shaping policy [73]. As a result of these challenges, it is crucial to implement practical strategies that efficiently manage waste overflow while fostering sustainable development in food production.

## 8. LCA in Retail and Consumption: Waste Management and Recycling

LCAs are widely applied in retail and consumption to evaluate the environmental impacts of waste management and recycling strategies, providing critical insights for enhancing sustainability across supply chains [74]. The transition to a circular economy in food retail plays a crucial role in addressing waste management and promoting sustainable consumption practices. CE principles in food retail focus on minimizing food waste, enhancing recycling systems, and designing closed-loop supply chains to retain the value of materials for as long as possible [75]. Figure 2 illustrates the concept of a circular economy transition within the food retail sector, highlighting interconnected strategies such as waste prevention, redistribution, recycling, and the integration of innovative technologies to improve sustainability and reduce the carbon footprint. This framework guides stakeholders in aligning retail operations with circular economy goals while addressing the growing demand for environmental accountability.

The environmental footprint of the food retail sector is substantial, with energy use, refrigeration, and waste management posing significant challenges. The Global Agenda 2030 program encourages countries to reassess their production, waste management, and resource reuse models to promote sustainable practices [76]. Mitigating this environmental impact requires the adoption of energy-efficient practices and technologies. Integrating advanced technologies throughout the supply chain is becoming increasingly essential as the production cycle shifts from a linear to a circular model [77,78]. Applying LCAs in the retail and consumer sectors is vital for improving waste management and recycling practices, particularly since plastic packaging accounts for nearly 40% of the global plastic demand [79].

In the food retail sector, key focus areas include energy use, refrigeration systems, and waste management practices, all of which contribute significantly to the overall carbon footprint and resource consumption. Additionally, consumer-level impacts such as food waste, energy consumption, and packaging disposal play a pivotal role in shaping the sustainability landscape of food retail [80]. Energy consumption in food retail establishments is a significant contributor to their environmental footprint, primarily driven by the use of lighting, heating, cooling, and refrigeration systems. Refrigeration is essential for preserving perishable goods, but is also one of the most energy-intensive components of food retail operations [81]. Reducing the environmental impact of this industry requires a shift towards sustainable solutions. The choice of refrigerants is critical in determining the environmental impact, as traditional hydrofluorocarbons (HFCs) have high global warming potentials [82]. Transitioning to natural refrigerants, such as ammonia or carbon dioxide, can significantly reduce greenhouse gas emissions [83]. Food retailers can effectively reduce their carbon footprint and enhance the overall sustainability by addressing the technological and operational aspects of energy consumption.

Furthermore, implementing energy-efficient technologies, such as LED lighting, smart thermostats, and advanced insulation materials, can yield substantial energy savings [84]. Such approaches align with the goals of a circular bioeconomy, where resources are continually cycled back into production rather than disposed of as waste. Sustainable technology, logistics, and advisory services are critical to facilitate this transition. By implementing these strategies, retailers can optimize packaging choices, mitigate greenhouse gas emissions, and respond to increasing consumer demand for sustainable products. By doing so, the retail industry can play a pivotal role in promoting more sustainable consumption patterns, ultimately contributing to greater environmental and economic resilience.

Consumer behavior is a critical determinant of food retail’s environmental impact, particularly in food waste, energy use, and packaging disposal [85]. Food waste at the consumer level represents a loss of valuable resources and contributes to methane emissions when organic waste decomposes in landfills [86]. Packaging materials, often single-used and non-recyclable, exacerbate the waste management challenges retailers and municipalities face. Studies and reports show that the global average recycling rate for materials is approximately 40%, highlighting a critical need for improvement [87]. Plastic-based multilayer packaging (PMP) is gaining popularity in the food industry due to its strong protective qualities, which help extend the shelf life and reduce food waste [88]. By layering different types of plastic, PMP offers enhanced durability, making it a cost-effective solution for producers looking to cut packaging costs without compromising the product quality [89]. However, while PMP’s structure effectively shields food from spoilage, it also complicates recycling efforts. The combination of multiple plastic layers makes it difficult to separate materials, posing a challenge for recycling processes and increasing its environmental footprint. Through an LCA, there is a need to assess the overall impact of PMP better and explore alternative packaging options that balance food protection with recyclability, ultimately supporting more sustainable and circular waste management practices.

Figure 3 presents a conceptual framework for integrating circular economy principles into food retail operations. This framework emphasizes strategies such as adopting recyclable and biodegradable packaging, implementing innovative waste management systems, establishing closed-loop supply chains, and conducting consumer education initiatives to reduce food and packaging waste. It also highlights the role of digital technologies in enabling circular practices.

This process involves reintroducing materials used in production back into the supply chain, either as raw materials, by breaking down products, or through direct reuse [87]. As raw materials become increasingly scarce, circularity is gaining significance, necessitating the reuse of resources to ensure sustainability [90]. To mitigate these issues, adopting circular economy principles offers a promising pathway. Strategies such as food recovery programs, where surplus food is redistributed to those in need, and upcycling initiatives that transform waste into new products can significantly reduce the volume of waste generated [91].

Additionally, promoting reusable and recyclable packaging through consumer education and incentives can help close the material loop. Engaging consumers through awareness campaigns and providing clear information on sustainable practices can foster more responsible consumption patterns. By aligning consumer behavior with circular economy objectives, food retail systems can achieve greater resource efficiency and minimize their environmental footprint

## 9. LCA of End-of-Life: Food Waste, Recycling, and Resource Recovery

Managing food waste is a significant challenge in creating sustainable food systems, with important environmental and resource implications [92]. Commonly employed methods such as landfilling, composting, and anaerobic digestion have unique ecological trade-offs. Landfilling remains widespread but poses critical environmental issues, mainly through methane emissions generated by the anaerobic decomposition of organic waste [93,94]. Methane, a greenhouse gas far more potent than carbon dioxide, significantly exacerbates climate change, while leachate from landfills creates additional environmental risks. These limitations underscore the urgent need to prioritize alternative waste management strategies [95].

Composting, by contrast, offers an opportunity to transform food waste into nutrient-rich organic matter that supports soil health and reduces reliance on synthetic fertilizers [96]. However, its environmental performance depends on variables such as energy use, emissions during the composting process, and the logistics of waste transportation. Moreover, the success of composting initiatives hinges on well-segregated waste streams, a challenge in many urban contexts [97,98].

Anaerobic digestion offers a more comprehensive approach by addressing both waste treatment and energy recovery [99]. Organic matter is converted into biogas and digestate through microbial activity in oxygen-deprived conditions. Biogas, primarily a mixture of methane and carbon dioxide, serves as a renewable energy source for electricity, heating, or transportation fuel, thereby reducing dependence on fossil fuels [100]. Digestate, rich in nutrients, can be used as a soil amendment, closing the nutrient loop and contributing to circular agricultural systems. LCA analyses consistently demonstrate that anaerobic digestion has a lower global warming potential than landfilling and can achieve a net energy gain if biogas capture and utilization are optimized [101]. These advantages position anaerobic digestion as a cornerstone technology in circular food systems, striking a balance between waste reduction and the recovery of energy and resources.

Beyond traditional methods, innovative recovery technologies drive further progress in circular systems by enhancing nutrient recycling and energy efficiency. Techniques such as biochar production and struvite crystallization exemplify this shift [102]. Biochar sequesters carbon, improving soil fertility and water retention and making it a valuable tool in climate-resilient agriculture. Similarly, struvite recovery offers an effective solution to the growing need for phosphorus recycling, a critical nutrient for crop production [95]. Combined with biogas and compost applications, these approaches create a more sustainable, interconnected system where resources are continually reused rather than discarded. From an LCA perspective, technologies that minimize energy use while maximizing resource recovery demonstrate clear environmental benefits, aligning with the principles of a circular economy [94].

Transitioning toward zero-waste food systems requires coordinated efforts across the entire value chain, from production to consumption. Reducing post-harvest losses through improved harvesting techniques and enhanced storage infrastructure can significantly minimize waste on farms. In retail and household contexts, redistributing surplus food, fostering responsible consumption behaviors, and implementing public awareness campaigns can further mitigate waste. Technological innovations, including artificial intelligence (AI) and blockchain, are also reshaping supply chains by optimizing inventory management and reducing inefficiencies [98]. When assessed through an LCA framework, strategies aimed at minimizing waste generation at its source consistently yield the most significant reductions in environmental impact [96].

Transitioning to a circular economy in food systems needs industry-specific strategies that address distinct waste streams and resource recovery opportunities. The wine and olive oil industries are illustrative examples due to the significant organic by-products generated during their production processes. These residues can pose substantial environmental risks if not adequately managed, including water contamination and greenhouse gas emissions [103]. However, applying LCA methodologies enables the revalorization of these by-products into valuable resources, fostering waste reduction, renewable energy generation, and soil enhancement. Examining how circular economy principles have been operationalized in these sectors provides critical insights into the practical implementation of sustainable food production systems.

The wine and olive oil industries exemplify how circular economy strategies can be integrated into food production systems to manage by-products while reducing environmental impacts. For instance, in the wine industry, grape pomace—comprising skins, seeds, and stems left after pressing—represents a significant waste stream with a high potential for valorization. Grape pomace can be composted into organic fertilizers, enhancing soil health and reducing reliance on chemical inputs [104]. Additionally, biochar produced from grape residues has been utilized to improve the soil structure and sequester carbon, providing dual benefits of waste reduction and climate mitigation [102]. Moreover, extracting polyphenols and antioxidants from grape pomace for use in food supplements and cosmetics creates valuable by-products, highlighting the economic potential of waste management in the wine sector [105].

Similarly, the olive oil industry faces significant waste management challenges due to the large volumes of olive pomace and wastewater generated during processing. Traditionally considered environmental liabilities due to their high organic content and phytotoxic compounds [106,107], these by-products have gained renewed interest through innovations in the circular economy. For example, the anaerobic digestion of olive mill wastewater mitigates water pollution and produces biogas, contributing to renewable energy generation [100]. Additionally, olive pomace can be composted or processed into bioenergy feedstock, supporting waste reduction and energy recovery [95]. Furthermore, emerging technologies such as hydrothermal carbonization enable the conversion of olive residues into biochar, thereby improving the soil fertility and water retention, particularly in arid regions [98].

The adoption of these strategies reflects a growing commitment to transitioning from linear to circular food systems. Biogas production, composting, and nutrient recycling play a central role in reducing waste while recovering valuable resources. At the same time, zero-waste initiatives underscore the importance of addressing waste prevention holistically, from farm-level practices to consumer behavior. The further integration of advanced analytical tools with LCA methodologies is necessary to realize these benefits fully.

## 10. Challenges in Applying LCA to Food Systems from a Circular Economy Perspective

Applying an LCA to food systems presents several challenges, including data gaps and the complexities of assessing agricultural production and consumer behavior. One key issue is defining the boundary between the technological system and nature, as agricultural production is inherently part of the environment. Ideally, all crops in a rotation system should be considered, as others can influence one crop, and environmental impacts must be allocated accordingly [10]. Additionally, handling co-products such as straw and manure presents allocation challenges. Models are also needed to estimate nutrient and pesticide leakage, as well as to develop human toxicity and ecotoxicity assessments further. Moreover, consumer behavior, particularly shopping habits and household waste, has a significant impact on the overall environmental impact, with losses in the household phase accounting for a notable portion of the total impact.

Other challenges of LCAs in food systems within a circular economy framework are related to the data availability, quality, and consistency [108]. Food systems are inherently complex, involving diverse production practices, supply chains, and waste management methods. Obtaining reliable data across these stages is often difficult due to varying record-keeping practices, regional differences, and the lack of standardized reporting protocols [109]. For example, small-scale producers may require additional resources or technical expertise to collect precise environmental data, resulting in gaps that affect the overall accuracy of LCA studies [110]. Furthermore, data inconsistencies between regions and production systems can hinder meaningful comparisons and limit the transferability of findings.

The quality of the data used in LCA studies is equally critical, as incomplete or outdated datasets can compromise the reliability of environmental impact assessments [111]. In the context of food systems, where environmental outcomes depend on dynamic variables such as climate conditions, soil health, and farming techniques, ensuring data relevance becomes even more challenging. Moreover, differences in data collection methods can introduce biases, making it difficult to achieve consistent and comparable results. Addressing these issues requires greater collaboration among stakeholders, including researchers, industry representatives, and policymakers, to establish shared data standards and improve transparency across food value chains [112]. Despite its strengths, the LCA has inherent limitations in capturing the full spectrum of environmental and social impacts within food systems. Conventional LCA models often focus on quantifiable metrics such as greenhouse gas emissions, energy use, and water consumption [113]. However, qualitative factors such as biodiversity loss, ecosystem resilience, and social equity are more challenging to measure and are frequently omitted [114]. For example, assessing the social implications of land-use changes, such as the displacement of communities or the loss of cultural heritage, requires broader socio-economic analyses beyond typical LCA frameworks [115]. As a result, LCA findings may offer a limited perspective, underscoring the need for integrated assessment tools that incorporate both environmental and social dimensions.

Another significant challenge lies in balancing economic, environmental, and social goals within food systems. Circular economy principles emphasize resource efficiency and environmental sustainability, but these objectives can sometimes conflict with economic and social priorities [116]. For instance, adopting advanced waste management technologies, such as anaerobic digestion or biochar production, can yield environmental benefits, but may involve high initial costs, limiting their feasibility for small-scale producers [117]. Similarly, reducing food waste through strict quality standards can unintentionally exclude lower-income consumers from access to affordable food, highlighting the trade-offs between sustainability and social equity.

Moreover, the economic viability of circular practices often depends on policy incentives, market conditions, and consumer behavior [118]. Without supportive policies or financial mechanisms, businesses may struggle to adopt sustainable technologies or transition to more circular models [119]. This includes phasing out subsidies for environmentally harmful products and redirecting funds toward sustainable initiatives. In contrast, green bonds and sustainable financing mobilize capital for projects like renewable energy and waste management, with support from institutions like the World Bank [120]. Market-based instruments such as pollution taxes and emissions trading incentivize circular practices by internalizing environmental costs. Environmental, social, and governance (ESG) investments, along with tailored financial tools such as leasing and pay-per-use models, further promote the transition to circular business models. These mechanisms, often bolstered by public–private partnerships, establish a robust framework for fostering a resilient and resource-efficient economy [121]. Additionally, the consumer demand for environmentally friendly products must align with industry efforts to create a sustainable food system [122]. Bridging this gap requires well-designed policies that strike a balance between environmental sustainability, economic growth, and social welfare. To address these complexities, integrating multi-criteria assessment frameworks into LCA can provide a more comprehensive evaluation of food systems. Such frameworks can simultaneously consider economic costs, environmental impacts, and social equity, offering a more balanced perspective on sustainability trade-offs [111]. Additionally, participatory approaches that involve stakeholders from across the food system can enhance the legitimacy and relevance of LCA findings, fostering more inclusive and equitable policy development.

In conclusion, while an LCA remains a valuable tool for assessing environmental impacts in food systems, its effectiveness within a circular economy perspective is constrained by data challenges, methodological limitations, and competing sustainability goals. Advancing LCA methodologies by incorporating broader environmental and social indicators, enhancing data transparency, and promoting multi-stakeholder collaboration is crucial for supporting more sustainable, resilient, and inclusive food systems.

## 11. Policy Implications: Supporting Circular Economy Through LCA in Food Production

Governments play a pivotal role in promoting LCA and circular economy practices within food production systems by establishing supportive policy frameworks. Transparency, credibility, and public engagement are essential for the long-term acceptance and effectiveness of LCA tools [5]. Policy measures can mandate sustainability reporting, enforce environmental standards, and incentivize the adoption of eco-friendly technologies [78]. By integrating LCA requirements into national sustainability strategies, policymakers can create a regulatory environment that encourages transparency and continuous improvement. For instance, requiring companies to disclose the environmental impact of their products based on LCA results can enable businesses to adopt more sustainable practices while informing consumers about environmentally responsible choices. Incentivizing farmers, producers, and retailers is crucial for promoting the widespread adoption of LCA-driven practices [123]. Financial incentives such as tax breaks, subsidies for sustainable technologies, and grants for environmental research can reduce the economic burden of implementing LCA processes [124]. Additionally, direct payments for environmental services—such as carbon sequestration or biodiversity conservation—can reward agricultural practices that contribute to circular food systems [125]. For example, subsidizing composting infrastructure or anaerobic digestion plants can help farmers manage organic waste sustainably while reducing greenhouse gas emissions.

Retailers also play a crucial role in advancing circular economy goals through LCA-driven decision making [126]. Policies encourage sustainable procurement, reducing packaging waste, and supporting local sourcing to align retailers’ business models with circular economy principles. Governments can further support these efforts by offering incentives such as reduced import tariffs on eco-friendly products or grants for green supply chain innovations. Retailers that adopt sustainability certification programs based on LCA findings can gain a competitive advantage while contributing to broader sustainability targets. International agreements and regional policy frameworks have emerged to align food production systems with the goals of a circular economy [127]. The European Green Deal, for instance, emphasizes reducing food waste, promoting sustainable agriculture, and enhancing the resource efficiency through LCA-based assessments. Similarly, the United Nations’ Sustainable Development Goals (SDGs), particularly SDG 12 on responsible consumption and production, provide a global policy framework that supports circular economy initiatives [128]. These agreements foster cross-border cooperation and knowledge sharing, enabling countries to harmonize their sustainability efforts.

Regional policies tailored to specific agricultural and food production contexts can further strengthen LCA integration [129]. For example, the European Union’s Farm to Fork Strategy emphasizes the importance of an LCA in evaluating the environmental impacts of food systems while promoting sustainable food production practices. National policies such as carbon pricing mechanisms and environmental certification schemes encourage producers to adopt circular economy models [130]. In developing regions, international aid programs can support local capacity-building initiatives for LCAs, enabling local producers to adopt more sustainable production methods. In conclusion, supportive policy frameworks are crucial for mainstreaming LCAs and circular economy practices throughout the food production value chain. By combining regulatory measures with targeted incentives, policymakers can accelerate the transition toward more sustainable, resource-efficient, and environmentally responsible food systems [131]. Aligning national policies with global sustainability frameworks ensures that efforts toward a circular economy remain coordinated and impactful, fostering a more sustainable future for food production worldwide.

## 12. Future Perspectives and Innovations in LCA for Sustainable Food Systems

Future perspectives and innovations in LCAs for sustainable food systems focus on enhancing their accuracy, inclusivity, and real-world applicability [132]. As food systems face increasing pressure to meet sustainability targets, advancements in LCA methodologies emphasize the importance of dynamic modeling, region-specific data integration, and the consideration of social and economic dimensions alongside environmental impacts [133]. Emerging tools and technologies, such as blockchain and AI, enable the more precise tracking of resources and emissions across complex food supply chains [134]. These innovations enhance the granularity of sustainability assessments, supporting transparent decision making. Incorporating digital product passports (DPPs) into these frameworks can further bolster the traceability and accountability, creating a comprehensive foundation for sustainable food systems [135].

DPPs are becoming a game-changer in public policy, driving forward sustainability goals. These passports act as detailed digital records, providing standardized and verifiable information about a product’s entire lifecycle—from raw material extraction to recycling at the end of its life [136]. The concept aligns seamlessly with the European Union’s Circular Economy Action Plan, emphasizing transparency and accountability in resource use [137]. By integrating DPPs into policy frameworks, governments can use LCA methodologies to evaluate products based on sustainability indices [138]. This approach enables data-driven decision making while advancing the principles of a circular economy.

Moreover, DPPs provide a robust data infrastructure that complements LCAs by enabling the more accurate tracking of environmental impacts across a product’s life cycle [139]. They also facilitate the development of a Sustainability Index for products, empowering consumers and stakeholders to make environmentally conscious choices [140]. The DPPs enhance circularity by improving the supply chain traceability, supporting the eco-design by identifying key areas for improvement in product lifecycles, and ensuring compliance with evolving sustainability standards. Similarly, AI has transformed the implementation and accuracy of LCA methodologies. Its potential spans data collection, analysis, predictive modeling, and decision support, offering a comprehensive approach to evaluating the environmental performance [141]. These innovations strengthen the foundation for achieving a circular economy by bridging data gaps, optimizing resource use, and fostering sustainable practices across industries. Table 4 provides an overview of AI applications in LCAs, categorized by their functionalities and use cases.

## 13. Conclusions

This study highlights the significant role of LCAs in advancing sustainable food systems by providing a detailed evaluation of environmental impacts across the entire food supply chain. Through this analysis, an LCA effectively identifies critical hotspots in resource use, waste generation, and greenhouse gas emissions. Integrating circular economy principles, such as renewable energy adoption, precision agriculture, and waste valorization strategies, enhances sustainability and promotes low-carbon food systems. Furthermore, this integration enables the reuse of by-products, optimizes the resource efficiency, and reduces the dependency on finite resources.

Despite its strengths, LCA methodologies face several challenges, including data gaps, the limited consideration of social and regional factors, and inconsistencies in evaluating long-term sustainability impacts. Addressing these issues requires refining LCA frameworks to incorporate dynamic modeling, region-specific analyses, and broader socio-economic dimensions.

This review was limited to the peer-reviewed literature published in English, which may have introduced a language bias. Relevant studies published in languages other than English, particularly those documenting local or regional applications of LCAs and circular economy principles, may have been excluded. This limitation can skew the geographic representation of case studies and under-represent practices in non-English-speaking countries, thereby limiting the global applicability of conclusions. Given the rapid development of environmental assessment tools and circular economy strategies, the temporal cut-off for included studies may have excluded recent advances in LCA modeling, such as dynamic, prospective, and spatially explicit LCA frameworks. The omission of these evolving approaches may result in an under-representation of cutting-edge methodologies that better capture temporal variability, non-linear feedback, or regional heterogeneity in agri-food systems.

The exclusion of the grey literature (e.g., industry reports, government documents, and NGO assessments) may have resulted in the omission of practice-oriented studies or innovative pilot projects that have not yet been published in academic journals. These sources often include high-resolution data and context-specific strategies that complement academic research, especially in the food and beverage sectors. The included studies vary significantly in terms of system boundaries, functional units, allocation procedures, and impact assessment methods. This heterogeneity limits the ability to make direct comparisons or meta-analytical assessments. For instance, differences in whether post-consumer phases (e.g., retail, consumption, and end-of-life) are included can substantially affect the magnitude and location of environmental hotspots identified through an LCA.

Although this review emphasizes environmental LCAs, the integration of a Social Life-Cycle Assessment (SLCA) and Life-Cycle Costing (LCC) remains limited in the analyzed literature. Consequently, the review may not fully capture trade-offs or co-benefits across the environmental, economic, and social dimensions of sustainability. This omission is particularly relevant in the context of circular economy assessments, which are inherently multidimensional. Most of the reviewed studies rely on static LCA models with deterministic assumptions. Few incorporate sensitivity analysis, scenario modeling, or uncertainty quantification, which are crucial for robust decision support, particularly in contexts with high variability in agricultural inputs, climatic conditions, and consumer behavior. This limitation suggests that future research should adopt hybrid and probabilistic LCA approaches to support policy and investment decisions better.

Future research should focus on leveraging emerging technologies, such as AI and blockchain, to enhance traceability, data accuracy, and informed decision making in food systems. Policymakers, industry stakeholders, and researchers must collaborate to foster multi-criteria approaches that balance environmental, economic, and social priorities. By addressing these challenges and adopting innovative practices, LCA has the potential to drive the transition towards a more sustainable and circular food system, ensuring resilience and equity in meeting future food production demands.

## Figures and Tables

**Figure 1 foods-14-01553-f001:**
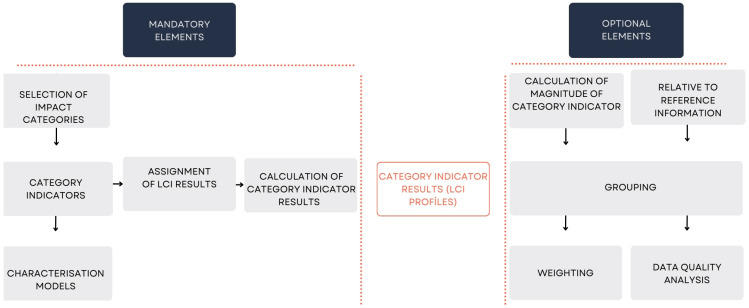
Elements of LCA (adapted from [10]). LCI stands for Life-Cycle Inventory.

**Figure 2 foods-14-01553-f002:**
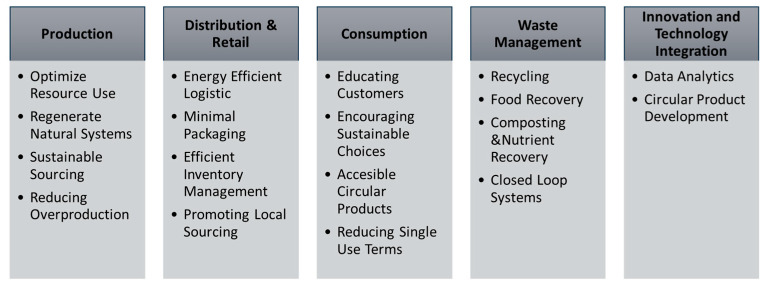
Circular economy transition in food retail.

**Figure 3 foods-14-01553-f003:**
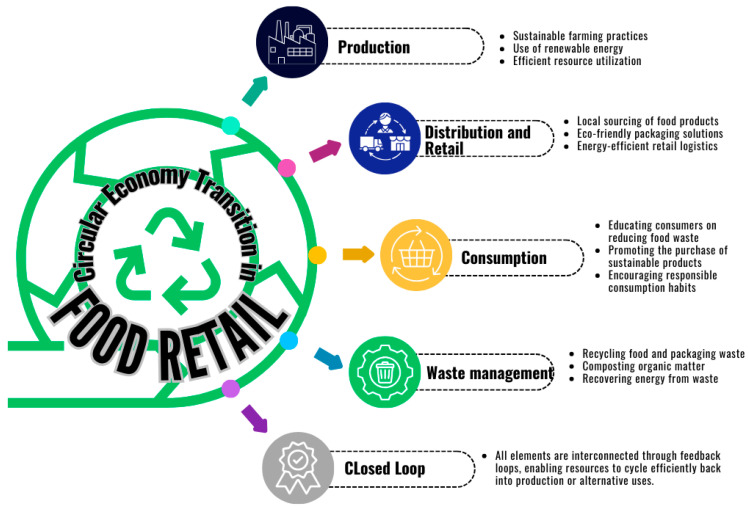
Circular economy in food retail.

**Table 1 foods-14-01553-t001:** Phases of the LCA framework based on the SETAC approach.

Phases	Description
Goal Definition and Scoping	Establishing assessment objectives, system boundaries, and functional units.
Inventory Analysis	Collecting and quantifying data on resource use, emissions, and waste throughout the life cycle.
Impact Assessment	Evaluation of environmental impact through three steps: classification, characterization, and valuation.
Improvement Analysis	Identifying strategies to reduce significant environmental impacts and implement sustainable practices.

**Table 2 foods-14-01553-t002:** Studies applying LCA, SLCA, and LCC in agri-food production.

Study	Method	Sector	Key Indicators Compared	Economic/Social Findings
[19]	LCC	Olive oil	Processing cost, energy recovery	Valorization reduces costs by ~25%
[18]	SLCA	Coffee, cocoa	Labor rights, child labor, wages	Social outcomes vary with scale
[12]	LCA	Chinese foods	Carbon, water footprint	Fresh vegetables lower GHG per kg

**Table 3 foods-14-01553-t003:** Key principles of the circular economy for sustainable food production.

Principle	Description	Ref.
Design Out Waste	Minimize waste throughout production by optimizing inputs, reducing losses, and reusing byproducts.	[21,22,23]
Keep Materials in Use	Recycle nutrients and materials into the production system to reduce dependency on new resources.	[24]
Regenerate Natural Systems	Focus on sustainable farming practices regenerating soil health, biodiversity, and ecosystems.	[25]
Shift to Renewable Inputs	Utilize renewable energy sources and biodegradable materials to minimize the environmental impact.	[22,26]
Encourage Nutrient Recovery	Implement strategies to recover nutrients (e.g., composting, anaerobic digestion) and reduce nutrient pollution.	[20]
Promote Biodiversity	Integrate diverse crops, livestock, and natural habitats to build resilience and sustainability in food systems.	[26,27]
Optimize Resource Efficiency	Maximize the efficiency of water, energy, and land use by utilizing precision farming techniques wherever possible.	[28]
Strengthen Local Supply Chains	Prioritize local sourcing and shorten supply chains to reduce emissions and promote regional economies.	[25]
Engage in Transparent and Responsible Practices	Ensure clear communication on production practices and engage with stakeholders to maintain accountability and transparency.	[29]
Encourage Circular Innovation	Invest in R&D for circular models, like upcycling food waste, closed-loop systems, and sustainable packaging alternatives.	[27,30]

**Table 4 foods-14-01553-t004:** Overview of AI applications in LCAs categorized by functionalities and use cases.

Application Type	Functionality	Use Cases	Ref.
Data Processing and Validation	Automates data cleaning and ensures accuracy	AI-based platforms standardize LCA datasets for consistency across industries.	[142]
Predictive Modeling	Anticipates environmental impacts	Predicts GHG emissions and resource use under various scenarios.	[118]
Ontology Creation	Develop frameworks for DPPs	AI creates digital frameworks integrating LCA and DPP for circular economy metrics.	[20]
Process Optimization	Improves resource efficiency	AI identifies hotspots in production processes, enabling targeted interventions.	[21]
Decision Support Systems	Guides sustainable policy making	Provides recommendations for material selection and end-of-life strategies based on LCA data.	[8]
Dynamic Impact Analysis	Real-time monitoring and feedback	Tracks live emissions data to adjust operational parameters dynamically.	[94]

## Data Availability

No new data were created or analyzed in this study. Data sharing is not applicable to this article.

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
