# Peer review of "Life-Cycle Assessment in Agri-Food Systems and the Wine Industry—A Circular Economy Perspective"

_foods, 2025, doi:10.3390/foods14091553_

Round 1

Reviewer 1 Report

Comments and Suggestions for Authors

Dear Authors,

The paper entitled "Life Cycle Assessment in Food and Alcoholic Drinks Production - A Circular Economy Perspective" is about agri-food and alcoholic beverage production in context a circular economy. The novelty inherent in the study, however, conceals a number of shortcomings:

1. The title should be revised because the paper almost never mentions "Alcoholic Drinks", but rather the wine industry. At the same time, issues related to the sustainability of the agri-food sector and the use of LCA for environmental impact analysis are considered. Perhaps this could be emphasized in the Title?
2. The abstract should be improved:
2.1. Specify the selection criteria, the period of the analysis and possibly which regions.
2.2. Specify what is the connection between food and alcoholic beverages and their combination in the analysis.
3. In the Introduction of the section, it is necessary to make a transition from production problems to distribution problems and then to possible solutions to look more complete.
4. In section 2, it would be desirable to disclose in more detail the research or models that will help improve the forecasting of the impact of LCA and the circular economy. Also, justify what barriers there are for different types of stakeholders in the implementation of the proposed circular economy model in the agri-food sector?
5. Despite the fact that the paper is a review, in section 3 the authors could provide a comparative analysis of statistics or other quantitative comparisons between different methods in the agri-food sector. For example, it would be interesting to update the economic indicators of social consequences.
6. It is also necessary to pay attention to what happens when scaling agricultural production, taking into account regional and technological differences in product processing. Indicate how circular economy technologies are used in this case. 7. The authors should pay attention to key studies on the implementation of circular economy principles, which should be added:
-Social Economy and the Transition Towards Circular Economy: a Survey Based Approach.
- Achievement of Sustainable Development Goals through the Implementation of Circular Economy and Developing Regional Cooperation.
- A Review of Circularity Indicators and Psychological Factors: a Comprehensive Analysis of Circularity Practices in Organizations
- Ensuring Sustainable Consumption Behaviours in Circular Economy Engagement.
- European Circular Economy Policy-Making in Sustainability and Resource Management Development.
8. The authors should delineate financial incentives that can be adapted for different sectors of the food industry. This also applies to the proposed Stimulation of the use of environmentally friendly technologies.
9. In section 12 "Final Remarks" it is worth changing to Conclusion and giving more detailed results of the analysis.
Minor:
1. There are typos and missing punctuation.
2. Some figure captions are broken and are located not under the figure, but after the text (Fig. 3).
3. The quality of Fig. 1 is poor and needs to be improved for readability.

Comments on the Quality of English Language

 The English should be polished.

Author Response

Dear Reviewer,

We sincerely appreciate your thoughtful feedback and constructive suggestions on our manuscript. We have carefully considered each of your comments and implemented the necessary revisions to enhance the clarity, accuracy, and scope of our work. Below is a point-by-point response detailing the changes made in the revised manuscript:

  1. We have revised the title better to reflect the content and focus of the paper. The new title is: “Life Cycle Assessment in Agri-Food Systems and the Wine Industry - A Circular Economy Perspective.”
  2. In response to your suggestion regarding the need to specify the selection criteria, analysis period, and the relationship between food and alcoholic beverages, we have made the following additions to the abstract: A systematic review of LCA applications in the agri-food sector was conducted, analyzing studies published across different regions. The selection criteria included relevance to circular economy strategies, waste valorization approaches, and assessing environmental impacts using LCA methodologies. The analysis explicitly explores the synergy between food and wine production within the broader agri-food system, considering shared sustainability challenges and opportunities for resource optimization.

These changes clarify the scope of the review, ensuring a stronger connection between food and wine production while providing transparency on our methodological approach.

  1. We have revised the introduction to highlight the transition from production challenges to distribution issues and then to potential solutions. Specifically, we have added a paragraph that connects the complexities of food production with distribution inefficiencies and the need for integrated solutions across the entire food system.
  2. In Section 2, we have expanded the discussion to provide more detail on the research and models that help improve the forecasting of the impact of LCA and the circular economy. Specifically, we now include references to system dynamics modeling and agent-based modeling, which facilitate a deeper understanding of the long-term environmental impacts and stakeholder behaviors associated with circular economy transitions.

Additionally, we have incorporated a new discussion on barriers faced by different stakeholders in implementing circular economy models in the agri-food sector. We highlight key challenges, including financial constraints, infrastructural limitations, a lack of awareness, and regulatory barriers, which can hinder the adoption of circular practices among both small and large producers.

  1. We thank the reviewer for this insightful suggestion. In response, we have expanded Section 3 to include a comparative summary of different LCA approaches applied within the agri-food sector. We have added examples where available, including updated references to recent economic indicators and social impacts, particularly from studies integrating Social Life Cycle Assessment (SLCA) and Life Cycle Costing (LCC).
  2. In response, we have expanded Section 4 (Farm-Level LCA: Inputs, Outputs, and Environmental Impact) to address the implications of scaling agricultural production, taking into account regional and technological differences. We now discuss how scaling affects environmental performance and highlight the role of circular economy technologies—such as anaerobic digestion and composting—that become more feasible and effective at larger scales. The revised text also reflects how regional infrastructure and policy contexts influence the successful implementation of these technologies.
  3. We agree that a solid grounding in the literature on circular economy implementation is essential. In our manuscript, we already address several key studies and frameworks
  4. We would like to clarify that the manuscript already addresses this point in Section 11, “Policy Implications: Supporting Circular Economy through LCA in Food Production.” Specifically, the text highlights the role of tailored financial instruments—such as subsidies for sustainable technologies, tax incentives, and grants—as key enablers of circular economy transitions. Moreover, sector-specific examples are discussed throughout the manuscript. For instance:
  • The wine industry is mentioned in the context of valorizing grape pomace through anaerobic digestion and renewable energy strategies (Section 6 and 9);
  • The dairy sector is referenced regarding the potential of precision agriculture and efficient feed practices (Section 4);
  • Composting, upcycling, and packaging innovations are discussed as circular solutions for small-scale processors (Sections 5 and 9).
  1. We appreciate the reviewer’s suggestion. In response, we have revised Section 12 and changed its title to “Conclusion” to better reflect the purpose and content of this final section. Additionally, we have expanded the section to include more detailed results of the analysis. Specifically, we now highlight how LCA identifies key environmental hotspots across the food supply chain and how the integration of circular economy principles contributes to resource efficiency and sustainability. The revised section also provides a forward-looking discussion on methodological limitations and directions for future research, thus strengthening the overall analytical depth and relevance of the conclusion.

Minor

  1. Every typo and punctuation mistake has been carefully polished throughout the manuscript, ensuring a clean and engaging read!
  2. We harmonized the location of Figure 3.
  3. The quality of Fig. 1 was improved.

Reviewer 2 Report

Comments and Suggestions for Authors

The paper is interesting in the topic explored but needs to be strongly improved, as for scientific rigour, first of all the main problem consists on the methodology, together with other content issues, in detail:
1. The paper does not clearly explain the methodology used for the literature review. It does not specify which databases were consulted (e.g., Scopus, Web of Science, Google Scholar), furthermore, there is no mention of the keywords used or the inclusion/exclusion criteria for selecting studies and it does not indicate the number of papers selected and reviewed. Without this information, it is difficult to assess the robustness of the review and its replicability.

  1. The paper mentions different Life Cycle Assessment (LCA) approaches, but it does not clearly define the system boundaries used in the reviewed studies. It is unclear which life cycle phases are included or excluded (e.g., does the study stop at the farm gate, or does it cover the full cycle from production to disposal?). Furthermore there is no critical discussion on how differences in system boundaries affect the comparability of studies.
  2. A comparative table summarizing methodological differences across the reviewed studies is missing, and it would be relevant to be added.
  3. While the paper emphasizes environmental sustainability, it largely ignores the social and economic dimensions and for example and Social Life Cycle Assessment (S-LCA)
  4. Finally the paper does not include a dedicated section on its limitations, reducing the transparency of the study. It does not discuss potential selection biases (e.g., exclusion of non-English studies, omission of the most recent research).

Author Response

Dear Reviewer,

We sincerely appreciate your thoughtful feedback and constructive suggestions on our manuscript. We have carefully considered each of your comments and implemented the necessary revisions to enhance the clarity, accuracy, and scope of our work. Below is a point-by-point response detailing the changes made in the revised manuscript:

  1. We have now added a dedicated Methodology section (section 2) to clarify the approach used for the literature review. This section outlines the databases consulted (Scopus, Web of Science, and Google Scholar), the keywords used for the search, and the inclusion and exclusion criteria applied to select relevant studies. Additionally, we have included information on the number of papers initially identified and those ultimately reviewed. This addition enhances the transparency, robustness, and replicability of our review.
  2. In the revised version of the manuscript, we have clarified the system boundaries used in the reviewed studies by adding “Most of the reviewed studies rely on static LCA models with deterministic assumptions. Few incorporate sensitivity analysis, scenario modeling, or uncertainty quantification, which are crucial for robust decision support, particularly in contexts with high variability in agricultural inputs, climatic conditions, and consumer behavior.”
  3. We agree that a comparative table could provide useful insights. However, given the extensive number of studies included in our review (143 references), such a table would become exceedingly long and potentially difficult to interpret in a concise and reader-friendly manner. For this reason, we opted to synthesize methodological differences through thematic grouping and narrative analysis within the manuscript.
  4. In response to your suggestion, we have now expanded the discussion to emphasize the importance of incorporating social and economic dimensions into the assessment of agricultural production systems. Specifically, we have added a section that highlights the role of Social Life Cycle Assessment (SLCA) in complementing environmental Life Cycle Assessment (LCA) by evaluating the impacts of food production on workers, local communities, and consumers.

To strengthen this discussion, we have incorporated recent literature. With these additions, the revised manuscript now offers a more comprehensive understanding of sustainability in the agri-food sector by integrating environmental, social, and economic assessments.

  1. Thank you for this important observation. In response, we have included a dedicated discussion of the study’s limitations in the newly revised “Conclusion” section. This addition enhances the transparency of our review. We now explicitly acknowledge that the analysis was limited to peer-reviewed literature published in English, which may have introduced language bias and led to the underrepresentation of studies from non-English-speaking regions. Furthermore, we address the exclusion of grey literature, the heterogeneity of methodological approaches in the included studies, and the limited incorporation of evolving LCA frameworks such as dynamic or spatially explicit models. Finally, we note the underrepresentation of socio-economic dimensions and the general lack of uncertainty analysis in the reviewed literature. This expanded reflection provides a clearer picture of the scope and boundaries of our work while outlining areas for future research.

Reviewer 3 Report

Comments and Suggestions for Authors

The review "Life Cycle Assessment in Food and Alcoholic Drinks Produc- 2
tion - A Circular Economy Perspective" is a very general work. Real applications of LCA were not included. Authors described general statements of LCA including different aspects, but without any significance. Examples of LCA with scientific data should have been included. No significant knowledge can be obtained from this review.

Author Response

We thank the reviewer for this critical feedback. We respectfully acknowledge the comment and would like to clarify that our intention with this review was not to provide an in-depth technical analysis of individual Life Cycle Assessment (LCA) case studies, but rather to offer a broad, descriptive overview of how LCA has been applied in the context of food and alcoholic beverages production within a Circular Economy framework. Our goal was to highlight patterns, identify knowledge gaps, and provide direction for future research rather than to replicate existing case-specific LCA evaluations.

To address this concern and enhance the value of the manuscript, we have revised the structure and added illustrative examples from selected studies that demonstrate LCA applications with quantitative or methodological relevance. We hope these improvements contribute to the clarity and significance of our review and better reflect the scientific contributions of the existing literature in this area.

Round 2

Reviewer 1 Report

Comments and Suggestions for Authors

The authors have worked out each comment in detail. The only thing is that there are too many sections (13), perhaps the authors could have combined and shortened the text of the paper a little. But overall, it is an great work.

Reviewer 2 Report

Comments and Suggestions for Authors

In my opinion the Author have done a good job in reviewing the paper, followingReviewer's suggestion, and I believe it is acceptable in the current form!

Reviewer 3 Report

Comments and Suggestions for Authors

The manuscript was seriously improved considering Reviewer's suggestions.